# Sensory Interaction and Balancing Ability Evaluation of the Elderly Using a Simplified Force Plate System

**DOI:** 10.3390/s22228883

**Published:** 2022-11-17

**Authors:** Jeong-Woo Seo, Joong Il Kim, Taehong Kim, Kyoung-Mi Jang, Youngjae Jeong, Jun-Hyeong Do

**Affiliations:** 1Digital Health Research Division, Korea Institute of Oriental Medicine, Daejeon 34054, Republic of Korea; 2Open XR Platform Convergence Research Center, National Research Council of Science and Technology, Daejeon 34141, Republic of Korea; 3KM Data Division, Korea Institute of Oriental Medicine, Daejeon 34054, Republic of Korea

**Keywords:** Wii balance board, simplified force plate system, sensory interaction, elderly, balancing ability

## Abstract

The Wii balance board (WBB) is a simplified force plate system used to evaluate the balancing ability of the elderly via a sensory interaction task to confirm a significant standing balance index. The accuracy of this system has been verified in previous studies. In this study, an instrumented and modified clinical test of sensory interaction on balance (i-mCTSIB) was performed on 84 elderly subjects, and the variables for center of pressure (CoP) were calculated using WBB for each task condition. The results indicate that the visual condition has a significant effect on the sway proprioceptive sensory variables with a foam condition as their complexity increases. In addition, the correlation between the variable and Berg Balance Scale was not confirmed since CTSIB is a sensory interaction on balance ability. Therefore, WBB can be used to evaluate balancing ability based on sensory interactions consisting of the surface condition.

## 1. Introduction

Balance is the most basic ability required to maintain body postures. To prevent falling while maintaining postures such as standing, walking, and exercising, the shape and strength of the muscles of each segment are controlled via various neural sensory adjustments to maintain balance [1]. However, balancing ability decreases with age due to decreased sensitivity of the nervous system and reduced muscle strength due to changes in the nervous system [2]. Therefore, accidents such as falls are common in the elderly.

Typically, visual, vestibular, and proprioceptive senses are used to maintain balance [3,4,5]. The contribution of each sensory component varies according to the complexity level of the posture. Several studies have been conducted to verify this. One study evaluated the change in sensory contribution measured via a force plate when performing a sensory organization test that varied visual input by tilting posture among different age groups [6]. Another study quantified and compared sway according to age when performing various postures [7]. Alternatively, another study confirmed the effect of sway on somatosensory feedback [8]. Additionally, maintaining balance when evaluating the balancing ability due to varying sensory input is necessary. Thus, a method is required to evaluate the balancing ability by quantifying the change in the center of pressure (CoP) to the ground while maintaining posture. The CoP applied to the ground is quantitatively measured by a force plate to identify and quantify balancing ability [9]. Two-dimensional directional values, i.e., the anterior-posterior (AP) and medial-lateral (ML) direction, of the force in contact with ground are recorded and calculated. The rate of change of CoP trajectory recorded while maintaining standing posture, change in distance, and total trajectory area can be used as indicators to evaluate the balancing ability. Various studies have been conducted to identify the variable for assessing the balancing ability using the CoP variables acquired via the force plate. Accordingly, a study reported potential markers for postural instability and fall risk in older adults [10], and another study evaluated the motor balance behavior using CoP mean velocity, permutation entropy, and detrended fluctuation analysis variables [11]. Thus, the possibility of effective balancing ability evaluation in the elderly has been verified.

Most force plates used to calculate CoP are relatively expensive as they are equipped with sensors and signal processing algorithms enabling precise and accurate measurement. Hence, evaluating balancing ability in a clinical or experimental environment is expensive. Moreover, monitoring the balancing ability is difficult for the elderly who have relatively low clinical access. The Nintendo Wii Balance Board (WBB) (Nintendo Inc., Kyoto, Japan) can be used to address this. It is a low-cost simplified force plate developed in October 2009 as an accessory for a home exercise game called Wii Fit Plus. Subsequently, it has been used in various balancing ability-related studies owing to the ease data acquisition via Bluetooth communication. Additionally, no significant difference in data has been observed compared with the actual expensive force plate. A systematic review of studies that assessed the reliability and validity of the WBB reported that it is a reliable and valid tool for assessing standing balance [12]. Moreover, several previous studies have reported no significant difference between the force plate and WBB readings. However, additional research is necessary to evaluate whether various balancing ability tests can be conducted, which is the aim of this study. A representative method known as the clinical test of sensory interaction on balance (CTSIB) is performed to measure balance maintenance time when maintaining each posture that poses a challenge to the vestibular and proprioceptive senses. To confirm the effect of sensory interaction, an instrumented method of measuring CoP in the CTSIB via WBB for the elderly has been proposed here. In this study, visual, vestibular, and proprioceptive senses affecting the balancing ability of the elderly were identified using the WBB. In addition, the correlation between the balancing ability evaluation index, the Berg Balance Scale (BBS), and the variables was confirmed. The purpose of this study is to confirm the possibility to quantifying sensory input and evaluating balancing ability in the elderly using WBB.

## 2. Materials and Methods

### 2.1. Data Acquisition System Configuration

A commercially available force plate, the Nintendo WBB (Nintendo, Kyoto, Japan), was used to measure postural control under various sensory conditions. Previous studies have verified the reproducibility of the WBB and its similarity with a force plate [12]. The dimensions of the WBB are 24.02 × 16.14 × 6.3 inches and has a wireless communication frequency of 2.4 GHz via Bluetooth version 1.2. The sampling interval and weight limit is 0.01 s and 136 kg, respectively. A strain gauge-based load sensor is installed at four positions: top left (TL), top right (TR), bottom left (BL), and bottom right (BR).

Typically, the WBB is operated by connecting it to the Nintendo Wii game console system. However, transmitting data by connecting it to a laptop with a Bluetooth receiver is possible owing to the Bluetooth communication capabilities of the WBB. In this study, a Bluetooth version 1.2 receiver was attached to the laptop and to connect the WBB wirelessly, as shown in Figure 1. Additionally, the WBB provides a “Synchro” button for easy pairing with a Bluetooth receiver.

Data was received by the laptop in real time at a sampling frequency of 40 Hz using an open source code developed by the Neuromechanics Lab at the University of Colorado [13]. The script of this code is named “bb_record” (balance board record). MATLAB (Mathworks Inc., Natick, MA, USA) was used for data analysis as it can record and display the CoP acquired by the Wii Balance Board in real time, as shown in Figure 2.

### 2.2. Participants

A total of 84 subjects (60 female and 24 male) participated in this experiment. Subjects with no critical damage to the musculoskeletal system, capable of walking independently, and able to stand were selected. This study included participants with no cognitive impairment to ensure complete understanding of the experimental protocol. The exclusion criteria were cases with a history of diseases related to visual, vestibular, and proprioceptive functions that affect balance. Prior to the experiment, experimental precautions approved by the institutional review board were explained and informed consent was obtained from the participants (IRB No.7001355-201505-HR-057, Institutional Review Board of Konkuk University). The characteristics of subjects are as shown in Table 1.

### 2.3. Experimental Protocol

The BBS was implemented to confirm the clinical balancing ability criteria of the subjects. It consists of 14 items categorized in sitting, standing, and postural change, which are dynamic tasks related to balance [14]. The Korean mini mental state examination (K-MMSE) was performed considering the correlation between cognitive function and balance.

The instrumented and modified clinical test of sensory interaction on balance (i-mCTSIB) provides a clinician with a means to quantify postural control under various sensory conditions. Subjects are instructed to perform the following six sensory conditions: (1) stand on a firm surface with eyes open, (2) stand on a firm surface with eyes closed, (3) stand on a foam surface with eyes open, (4) stand on a foam surface with eyes closed, (5) stand on two foam surfaces with eyes open, and (6) stand on two foam surfaces with eyes closed. The vision of the participant was fixed on the target in front of them when open. The subjects are allowed to perform each sensory condition for 30 s with their arms crossed in front of the chest to maintain a standing posture. A sponge foam with dimensions of 40 × 40 × 7 cm was used, and two foams were used in the surface condition of foams. The sponge foam deformed when used, but restored to its original state after use. An eye patch was used during the visual off state, and a safety mat was placed to prevent injuries in case of a fall. Sufficient rest time was allowed between each stage of the experiment, and measurement was repeated thrice for each condition. The average was used to calculate the CoP.

### 2.4. Data Analysis

The measured values of force from the TL, TR, BL, and BR of the WBB were low-pass filtered at a threshold frequency of 10 Hz using a fifth order Butterworth filter to eliminate noise [15]. The following formula was used to calculate the CoP in both the directions of ML and AP:(1)$$\mathrm{CoP}\;\mathrm{ML}=\frac{L}{2}\;\frac{\left(\left(\;TR+BR\right)-\left(TL+BL\right)\right)}{TR+BR+TL+BL},\;\mathrm{CoP}\;\mathrm{AP}=\frac{L}{2}\;\frac{\left(\left(\;TR+TL\right)-\left(BR+BL\right)\right)}{TR+BR+TL+BL},\;L=\mathrm{length}\;\mathrm{of}\;\mathrm{the}\;\mathrm{WBB}\;(43.3\;\mathrm{cm})\;$$

The CoP range refers to the distance between the positions measured as the furthest distance from the center in the ML and AP directions. CoP covariance is the variance in each direction, and CoP velocity is the distance per second. 95% COV of the eigenvalue indicates 95% of the variance value when the axis is transformed in the eigenvector direction for each direction. The statistical differences between the visual on and off conditions based on the effect of surface were compared from the calculated variables. In addition, the correlation between each condition and the BBS score was verified, and the modified Romberg ratio was calculated using the average values for each condition to confirm its visual contribution to postural stability using Equation (2). A modified Romberg ratio close to zero or negative implies that visual information is less important for postural control [16].
(2)$$\mathrm{modified}\;\mathrm{Romberg}\;\mathrm{ratio}=\frac{Eye\;close-Eye\;open}{Eye\;close+Eye\;open}\times 100$$

## 3. Results

Figure 3 shows the changes in CoP for each standing posture condition. The figure highlights the difference in the distribution area of CoP according to the visual input and difficulty of surface condition.

The three surface conditions with eyes open, that is, a visual on state, represent a statistically significant difference between all conditions except Foam-1 and 2 for the CoP covariance AP and CoP range AP (Table 2).

Statistically significant difference was confirmed upon comparing the three surface conditions with eyes closed, which is the visual off state (Table 3).

We calculated the modified Romberg ratio to check the visual dependency of each condition. For the firm surface conditions, negative values were recorded for the CoP range and covariance ML, 95% COV of eigenvalue AP, and 95% ellipse sway area. CoP covariance ML in one-foam and two-foam conditions, showed positive values for all the variables (Table 4).

Similarly, correlation analysis between the BBS and variables of each condition indicated a significant correlation for 95% COV of eigenvalue ML and 95% ellipse sway area when eyes were closed for the one-foam condition. A significant correlation was confirmed for 95% COV of eigenvalue ML for the two-foam condition with eyes open as well (Table 5).

## 4. Discussion

In this study, we confirmed the possibility of evaluating balancing ability according to the sensory condition of the elderly using a representative WBB as a simplified force plate. The difference between the ten variables calculated from the CoP measured for each experimental condition of the CTSIB was confirmed.

The average value of the ellipse sway area, which indicates the overall area of sway generated when performing the standing posture according to the sensory condition, shows that sway was significantly small for the firm surface visual on condition. Accordingly, standing posture was easy to achieve. Further, the one-foam surface condition with eyes open had a larger sway area than that of eyes closed. Additionally, the vestibular and proprioceptive senses were observed to be more complex than vision, thus affecting the balancing ability and inducing more sway. The foam surface condition was confirmed to have been influenced by the presence or absence of vision. Additionally, the influence of vision was greater in the two-foam condition, and the complexity of proprioceptive senses was higher. Typically, in the eye-closed condition, distinguishing increased oscillation was very difficult as the subjects were either unstable or greatly confident in their ability to control oscillation [17]. Consequently, similar to a previous study that compared the CoP area when performing CTSIB, the eyes-closed two-foam condition had the highest complexity resulting in higher sway [16].

As a result, in Table 2, although the difficulty affecting vestibular and proprioceptive increased as the surface condition changed in the visual on condition, CoP covariance AP and CoP range AP as variables did not show a difference in CoP characteristics. This implies that when the difficulty of the vestibular and proprioceptive senses is slightly increased, the CoP range and variance in the AP direction are not significantly affected. However, a difference in values is recorded when the difficulty level is the least or highest.

In contrast, the results shown in Table 3 indicate a difference at all levels of difficulty for visual conflict. Combining the two results, a difference as large as firm and two-foams conditions based on the difficulty of vestibular and proprioceptive should be identified to confirm the difference according to the effect on sensory.

Therefore, during vision on or off, balancing ability is exhibited and is compensated for by the difficulty of a small surface condition. Therefore, configuring the task based on visual and surface conditions to evaluate the balancing ability is necessary. Additionally, results of previous studies on visual information and surface conditions show similar results [18]. The primary function of the vestibular-spinal tract (VST) is to maintain balance and that of the vestibular-ocular system (VOS) and reflex functions is to maintain visual stability during head movement. The pathways of the two neural functions, namely VST and VOS are not identical neuronal circuitry and hence work independently. In addition, VOS functionality is limited to the two-foam condition without vision and the head relatively shaken; therefore, maintaining balance with VST only is difficult.

Table 4 presents more specific visual information, thus confirming the modified Romberg ratio for each surface condition. The CoP range, covariance ML, 95% COV of eigenvalue AP, and 95% ellipse sway area in firm condition, and the CoP covariance ML in one-foam condition showed negative values. All the variables in two-foams condition were positive. However, visual information is not significant when the Romberg ratio is zero or negative [19]. Thus, visual information has an important influence on all variables for foam conditions. Two of the three variables of the firm condition are range and covariance in the ML direction, which are generally affected by vision in the AP direction, where maintaining balance by swaying is relatively easy. However, the influence of the presence or absence of vision is not significant in the ML direction as the forward and backward movements, rather than left and right, are controlled to maintain posture [20].

Various assessments are used to evaluate balancing ability in clinical practice. A representative example is the BBS. Significant correlations were confirmed for the 95% COV of eigenvalue ML, and 95% ellipse sway area in eyes closed and one-foam condition, and 95% ellipse sway area in eyes open and two-foam conditions. Previously, the balance ability was confirmed to be regulated by the influence of AP sway. Thus, performing additional experiments to confirm the cause for significant correlation with BBS is necessary. These experiments consist of various tasks such as sitting, transfer, arm reaching, turning, and one leg standing. Alternatively, CTSIB evaluates balancing ability particularly in terms of sensory organization. Therefore, the correlation between the two assessments is not considered to be high. i-mCTSIB would be more useful when evaluating only specialized balancing ability with a dependency on sensory organization and interaction. WBB would also be useful as a measuring instrument in place of an expensive force plate used in clinical practice. Further, the detailed dependency of each sensor can be quantified using CoP variables. Similar to the Romberg ratio confirming the dependence of visual functions on each CoP variable based on surface conditions, evaluating the individual dependence of vestibular and proprioceptive senses, deriving a more degraded sense of detail, and presenting diagnostic criteria for sensory function training is possible.

The limitations of this study have been presented further. The effect on balance ability due to gender was not considered. Future studies will aim to collect additional data and perform comparisons based on gender. In addition, the difficulty control according to foam is not sufficient for vestibular and proprioceptive sensory monitoring. In the experimental design of this study, vestibular and proprioceptive perturbation of slightly less difficulty was controlled using only sponge foam. Therefore, the discussion focused on visual on or off rather than vestibular and proprioceptive sensory input. In the future, we changes in CoP under conditions that can control the difficulty of proprioceptive senses in dynamic movement, body tilting, standing posture, and gait kinematic variable from center of mass by inertial measurement unit that can affect the actual vestibular will be examined [21].

## 5. Conclusions

In this study, we performed sensory interactions to evaluate balancing ability of the elderly using a Nintendo WBB, which is a relatively simple and easily accessible force plate. In addition, we confirmed that visual effect had a greater impact on sway as the proprioceptive sensory difficulty increased. In addition, in the presence of foam, which increases the difficulty of the surface condition, visual information is vital for all variables. Some variables were identified as CoP variables correlated with BBS, thereby confirming overall balancing ability; however, no significant correlation was confirmed since they were specialized for sensory interaction. Thus, sensory interaction-based balancing ability using a simplified system instead of expensive laboratory or clinical equipment was evaluated. Therefore, tasks specialized for evaluating balancing ability can be configured and for evaluation using WBB.

## Figures and Tables

**Figure 1 sensors-22-08883-f001:**
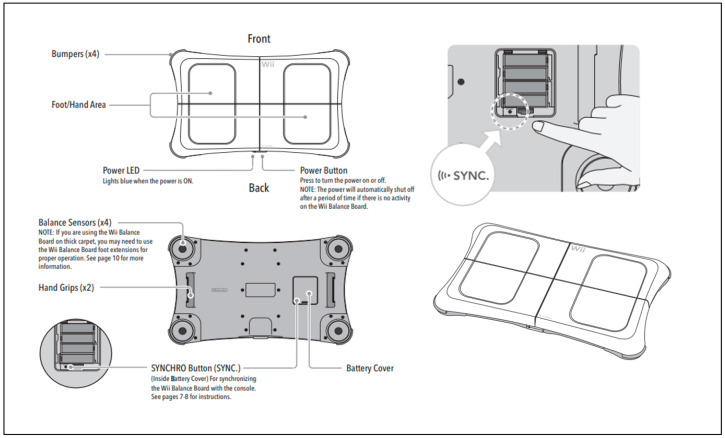
Components and structure of the WBB (from the Operations Manual).

**Figure 2 sensors-22-08883-f002:**
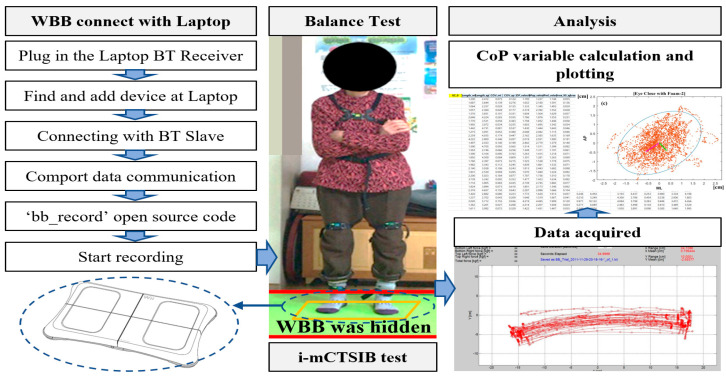
WBB connection and data acquisition algorithm.

**Figure 3 sensors-22-08883-f003:**
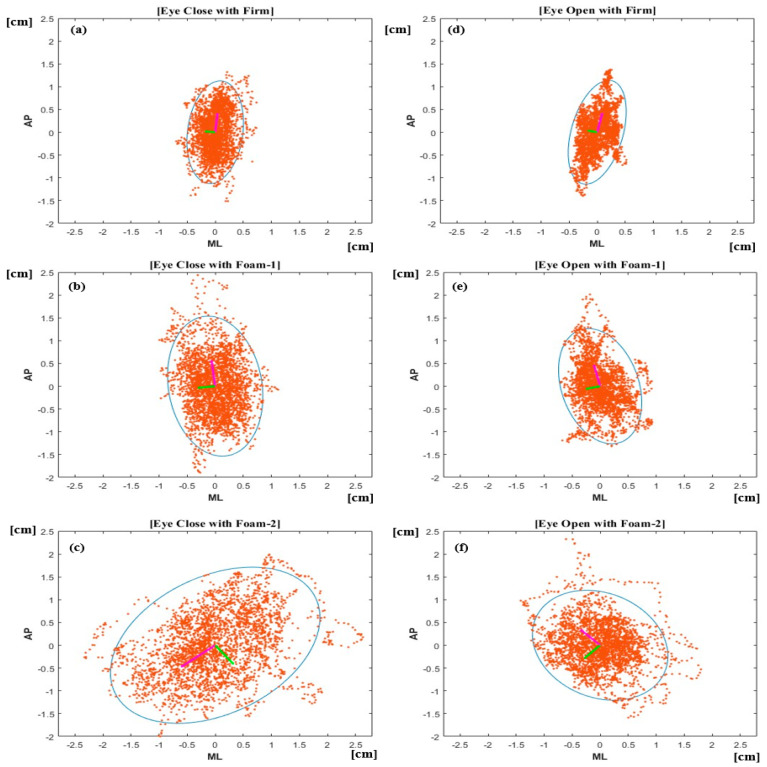
CoP for (**a**) eyes closed on firm ground, (**b**) eyes closed on Foam-1, (**c**) eyes closed on Foam-2, (**d**) eyes open on firm ground, (**e**) eyes open on Foam-1, and (**f**) eyes open on Foam-2. Blue circle: 95% covariance of eigenvalue area, Pink arrow: largest eigen vector, Green arrow: smallest eigen vector.

**Table 1 sensors-22-08883-t001:** Characteristics of the subjects.

Variable	Subjects		
	Female	Male	*p*-Value
No.	60	24	-
Age (years)	77.37 ± 4.99	79.83 ± 4.73	* 0.04
Height (cm)	149.57 ± 4.52	164.88 ± 6.29	* 0.00
Weight (kg)	56.07 ± 7.46	68.68 ± 10.22	* 0.00
Berg Balance Scale (score)	53.05 ± 1.83	53.08 ± 2.73	0.95
K-MMSE (score)	17.99 ± 3.47	22.66 ± 2.74	* 0.00

Independent *t*-test (* *p* < 0.05).

**Table 2 sensors-22-08883-t002:** Difference between the surfaces with normal visual condition.

Variables	Firm-Eye Open	Foam1-Eye Open	Foam2-Eye Open
CoP Range ML [cm]	2.14 ± 1.13 *^#^	2.83 ± 1.12 *^$^	4.29 ± 1.78 ^#$^
CoP Range AP [cm]	3.28 ± 0.94 *	3.50 ± 0.93	3.72 ± 1.11 *
CoP Covariance ML [cm]	0.14 ± 0.16 *^#^	0.29 ± 0.31 *^$^	0.54 ± 0.38 ^#$^
CoP Covariance AP [cm]	0.35 ± 0.21	0.36 ± 0.17	0.38 ± 0.26
CoP Velocity ML [cm/s]	1.46 ± 0.33 *^#^	1.79 ± 0.39 *^$^	2.14 ± 0.45 ^#$^
CoP Velocity AP [cm/s]	1.47 ± 0.38 *^#^	1.84 ± 0.47 *^$^	2.14 ± 0.57 ^#$^
CoP Velocity RMS [cm/s]	1.46 ± 0.23 *^#^	1.76 ± 0.31 *^$^	2.18 ± 0.40 ^#$^
95% COV of Eigenvalue AP [cm]	0.11 ± 0.10 *^#^	0.18 ± 0.11 *^$^	0.28 ± 0.13 ^#$^
95% COV of Eigenvalue ML [cm]	0.38 ± 0.23 *^#^	0.47 ± 0.31 *^$^	0.64 ± 0.43 ^#$^
95% Ellipse sway area [cm^2^]	11.65 ± 8.56 *^#^	16.15 ± 9.81 *^$^	26.18 ± 16.29 ^#$^

Mean ± Standard deviation, CoP: Center of Pressure, RMS: root mean square, COV: Covariance, Repeated. Measured ANOVA with Bonferroni multiple comparisons, significantly different: *^, #, $^
*p* < 0.05.

**Table 3 sensors-22-08883-t003:** Difference between the surfaces with closed visual condition.

Variables	Firm-Eye Close	Foam1-Eye Close	Foam2-Eye Close
CoP Range ML [cm]	1.97 ± 0.88 *^#^	2.96 ± 1.36 *^$^	4.98 ± 2.01 ^#$^
CoP Range AP [cm]	3.54 ± 1.19 *^#^	4.04 ± 1.02 *^$^	4.71 ± 1.49 ^#$^
CoP Covariance ML [cm]	0.13 ± 0.15 *^#^	0.26 ± 0.18 *^$^	0.77 ± 0.58 ^#$^
CoP Covariance AP [cm]	0.41 ± 0.29 *^#^	0.50 ± 0.26 *^$^	0.64 ± 0.38 ^#$^
CoP Velocity ML [cm/s]	1.77 ± 0.51 *^#^	2.21 ± 0.62 *^$^	2.69 ± 0.71 ^#$^
CoP Velocity AP [cm/s]	1.92 ± 0.69 *^#^	2.52 ± 0.86 *^$^	2.91 ± 0.90 ^#$^
CoP Velocity RMS [cm/s]	1.52 ± 0.23 *^#^	1.83 ± 0.35 *^$^	2.50 ± 0.62 ^#$^
95% COV of Eigenvalue AP [cm]	0.10 ± 0.09 *^#^	0.20 ± 0.14 *^$^	0.45 ± 0.28 ^#$^
95% COV of Eigenvalue ML [cm]	0.44 ± 0.31 *^#^	0.56 ± 0.27 *^$^	0.96 ± 0.60 ^#$^
95% Ellipse sway area [cm^2^]	11.58 ± 7.67 *^#^	19.60 ± 12.04 *^$^	38.71 ± 23.04 ^#$^

Mean ± Standard deviation, CoP: Center of Pressure, RMS: root mean square, COV: Covariance, Repeated; Measures ANOVA with Bonferroni multiple comparisons, significantly different: *^, #, $^
*p* < 0.05.

**Table 4 sensors-22-08883-t004:** Modified Romberg ratio for each surface condition.

Modified Romberg Ratio	Firm	Foam-1	Foam-2
CoP Range ML [cm]	−4.22	2.35	7.51
CoP Range AP [cm]	3.89	7.20	11.76
CoP Covariance ML [cm]	−4.67	−5.09	17.43
CoP Covariance AP [cm]	7.89	17.12	24.92
CoP Velocity ML [cm/s]	9.66	10.42	11.35
CoP Velocity AP [cm/s]	13.32	15.42	15.25
CoP Velocity RMS [cm/s]	2.33	2.11	6.95
95% COV of Eigenvalue AP [cm]	−5.29	7.06	22.71
95% COV of Eigenvalue ML [cm]	7.18	8.85	19.82
95% Ellipse sway area [cm^2^]	−0.29	9.64	19.32

**Table 5 sensors-22-08883-t005:** Correlation between BBS score and each condition variable.

	Firm		Foam-1		Foam-2	
Variables	Eye Close	Eye Open	Eye Close	Eye Open	Eye Close	Eye Open
CoP Range ML [cm]	0.01	−0.04	−0.09	−0.02	−0.14	0.01
CoP Range AP [cm]	0.02	−0.21	−0.33	−0.16	−0.15	−0.10
CoP Covariance ML [cm]	−0.03	−0.12	−0.13	−0.12	−0.17	−0.20
CoP Covariance AP [cm]	0.07	−0.15	−0.25	−0.05	−0.13	−0.13
CoP Velocity ML [cm/s]	−0.06	0.06	−0.15	−0.07	−0.06	−0.03
CoP Velocity AP [cm/s]	0.01	−0.08	−0.20	−0.06	−0.11	−0.11
CoP Velocity RMS [cm/s]	0.11	0.05	0.04	0.06	0.03	−0.08
95% COV of Eigenvalue AP [cm]	−0.07	−0.08	−0.16	−0.10	−0.14	−0.12
95% COV of Eigenvalue ML [cm]	0.06	−0.18	−0.21 *	−0.09	−0.19	−0.21 *
95% Ellipse sway area [cm^2^]	0.01	−0.10	−0.21 *	−0.08	−0.19	−0.02

Correlation coefficient by Pearson’s r, significantly different: * *p* < 0.05.

## Data Availability

The data presented in this study are available upon request from the corresponding author.

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
