# Peer review of "Sensory Interaction and Balancing Ability Evaluation of the Elderly Using a Simplified Force Plate System"

_sensors, 2022, doi:10.3390/s22228883_

Round 1
Reviewer 1 Report
Dear Authors,
thank you very much for sending the article titled: "Sensory Interaction and Evaluation of the Balancing Ability of Elderly using a Simplified Force Plate System" for review. The approach of the study appears very original. The contents of the manuscript are quite interesting by his methodology and the tools used. Below are suggestions to the authors:
- Table 1. Please include p-value's between groups
- lack of inclusion and exclusion participant in experiment
- authors used Wii balance board (WBB) as a simplified force plate system for experiment. Have the authors considered the use of IMU sensors in research in the future? Installing sensors in different parts of the body (lower limbs, torso, head) would allow for obtaining additional parameters related to calving not only CoP but also individual body segments. For example, in the article entitled: A kinematic model of a humanoid lower limb exoskeleton with Pneumatic Actuators, Acta od Bioengineering and Biomechanics authors used IMU for obtain gait kinematic parameters. In my opinion IMUs will allow to obtain interesting parameters. To improve article quality please cite above manuscript and write a few sentences in Conclusion section about future research.
- lack of limitation of the study.
Author Response
Response to Reviewer 1 Comments
Thank you for your review comments on this manuscript. I think it helped a lot to improve the completeness of this manuscript by your review comment. I actively responded to the lack of content given by the reviewer and tried to apply it to the manuscript. Once again, thank you for reviewing this manuscript, and please check the following for responses to comments.
Point 1: Table 1. Please include p-value's between groups.
Response 1: According to the comments, Table 1. includes p-values between groups (independent T-test). Thank you.
Point 2: Lack of inclusion and exclusion participant in experiment
Response 2: The inclusion and exclusion participant in experimental criteria was added to 2.2. Participants. The contents are as follows. “Subjects with no critical damage to the musculoskeletal system, capable of walking independently, and able to perform standing were selected. In addition, this study included cases where there was no cognitive impairment enough to fully understand the experimental protocol. Exclusion criteria were cases where there was a history of diseases related to visual, vestibular, and proprioceptive functions that affect balance ability.” Thank you.
Point 3 : Authors used Wii balance board (WBB) as a simplified force plate system for experiment. Have the authors considered the use of IMU sensors in research in the future? Installing sensors in different parts of the body (lower limbs, torso, head) would allow for obtaining additional parameters related to calving not only CoP but also individual body segments. For example, in the article entitled: A kinematic model of a humanoid lower limb exoskeleton with Pneumatic Actuators, Acta od Bioengineering and Biomechanics authors used IMU for obtain gait kinematic parameters. In my opinion IMUs will allow to obtain interesting parameters. To improve article quality please cite above manuscript and write a few sentences in Conclusion section about future research.
Response 3: In future study, kinematic variable using CoM of IMU will be considered. Thank you for recommending a good reference. This reference (ref.[21]) has been added to the limitations of the last part of the discovery and to the future study plan. Thank you.
Point 4 : lack of limitation of the study.
Response 4: The limitations of this study are that the gender effect is not considered and the vestibular and proprioceptive perturbation using sponge foam is rather weak. This was added to the last part of the discussion (lines 253-262). Thank you.
Reviewer 2 Report
The paper is interesting for the research field of the Sensory Interaction and Evaluation of the Balancing Ability of 2 Elderly using a Simplified Force Plate System.
The paper is well written, correctly organized, with appropriate title, abstract, introduction, notations, content, conclusions and references. The current stage must be more exhaustive, critical, with presentations of the approaches of other authors, the results obtained by them, with clear references to the specialized literature published in this field.
The Discussions chapter presents the results and comparative analyzes regarding the way in which the visual effect had a greater impact on the sway as the proprioceptive sensory difficulty increased. The figures, tables and mathematical expressions in the paper seem correct from a scientific and technical point of view. The paper is correctly written in international English.
Author Response
Response to Reviewer 2 Comments
Thank you for your review comments on this manuscript. I think it helped a lot to improve the completeness of this manuscript by your review comment. I actively responded to the lack of content given by the reviewer and tried to apply it to the manuscript. Once again, thank you for reviewing this manuscript, and please check the following for responses to comments.
Point 1: The paper is well written, correctly organized, with appropriate title, abstract, introduction, notations, content, conclusions and references. The current stage must be more exhaustive, critical, with presentations of the approaches of other authors, the results obtained by them, with clear references to the specialized literature published in this field.
Response 1: Thanks for the good comments. Added missing references. Thank you.
Point 2: The iscussions chapter presents the results and comparative analyzes regarding the way in which the visual effect had a greater impact on the sway as the proprioceptive sensory difficulty increased. The figures, tables and mathematical expressions in the paper seem correct from a scientific and technical point of view. The paper is correctly written in international English.
Response 2: Thanks for the good comments. I have added what I think is a limitation to the discussion. Thank you.
Reviewer 3 Report
Measurement problems are not addressed: Foam alters the force signals.
The height of the foam alter CoP measure. The applicant force is referred to the surface of the platform while the force is applied on the foam surface that is until 14 cm higher.
The dynamic foam deformation (specific for the used materials) transfer the force on the force platform, its characteristics need to be described and measure. Furthermore, this make the experiment not reproducible if you do not use the same foam.
Is missing the main literature on the Sensory Organization as reported by the literature on SOT, see:
· Peterka RJ (2001) Sensorimotor integration in human postural control. J Neurophysiol 88(3):1097–1118. DOI: 10.1152/jn.2002.88.3.1097
· Nashner LM (2008) Practical Management for the Dizzy patient: Computerized Dynamic Posturography. Philadelphia, PA : Lippincott Williams & Wilkins.
In eye-closed condition, it is impossible to distinguish subject that increase oscillation because they are unstable or greatly confident on their ability in controlling the oscillation (the paper give ambiguous information):
· D'Anna, C., et al. (2015). Time to boundary function to assess upright stance in blind children. In Proceedings of the Annual International Conference of the IEEE Engineering in Medicine and Biology Society, EMBS (Vol. 2015-November, pp. 3468-3471). https://doi.org/10.1109/EMBC.2015.7319139
The vision in eye open condition is out of control; the task condition and the instruction alter the results because they alter the context of the task, also introducing cognitive aspects, see:
· Cappa P, et al. Effect of changing visual condition and frequency of horizontal oscillations on postural balance of standing healthy subjects. Gait Posture. 2008 Nov;28(4):615-26. doi: 10.1016/j.gaitpost.2008.04.013.
The pressure under the foot is under of control when you use the foam. The use of the term proprioceptive is too general in this case is preferred to refer to the pressure under the foot (the subjects simply stand).
The use of the term “tasks and variables” do not seem appropriate. The task is unique: standing. The contexts are changing (Solid surface, Foam1 and Foam2). One task and three contexts.
Without knowing the head and body displacements it is impossible to infers on the role of the vestibular system, please see other relevant references.
The perturbation induced by the alteration of the surface of support can drastically change the body frame of reference for maintain the equilibrium. In your experiment is out of control see:
· Amori V, et al. Upper body balance control strategy during continuous 3D postural perturbation in young adults. Gait Posture. 2015 Jan;41(1):19-25. doi: 10.1016/j.gaitpost.2014.08.003.
The paper need data integration and substantial revision: of the introduction integrating the proper literature, of the results integrating data, and of the discussion avoiding inference on the vestibular system on prorpiception and clearly defining all the limitations of this paper.
Author Response
Response to Reviewer 3 Comments
Thank you for your review comments on this manuscript. I think it helped a lot to improve the completeness of this manuscript by your review comment. I actively responded to the lack of content given by the reviewer and tried to apply it to the manuscrip. Once again, thank you for reviewing this manuscript, and please check the following for responses to comments.
Point 1:
Measurement problems are not addressed: Foam alters the force signals. The height of the foam alter CoP measure. The applicant force is referred to the surface of the platform while the force is applied on the foam surface that is until 14 cm higher.
Response 1:
It is correct that the Foam changes the force signals. However, I think the sensitivity of the sensor is enough to measure that. Even if there was a problem with measurement sensitivity, there were statistical differences in most variables under the condition of Foam 2. The purpose of this study is not to present absolute results value. In addition, the design of the experiment is the same as the experimental method called i-mCTSIB performed in numerous previous studies. In consideration of this, the commercialized measuring equipment (Biodex Balance system) is also performing CoP-based balance evaluation under the condition that the CTSIB indexed pad(sponge foam) is placed on the sensor. The foam used in this experiment is a sponge material, considering the height and material of the foam recommended in the traditional CTSIB experiment. Please understand it. Thank you.
Point 2:
The dynamic foam deformation (specific for the used materials) transfer the force on the force platform, its characteristics need to be described and measure. Furthermore, this make the experiment not reproducible if you do not use the same foam.
Response 2:
There was a misunderstanding because the material of the foam used in this experiment was not explained in detail. In this experiment, sponge foam was used, and it is a material that is deformed when used, but is restored to its original state when use is finished.(line 134-136). This is the same material used in several previous studies. The inability to give high perturbation to the vestibular and proprioceptive was suggested as a limitation. Thank you.
Point 3:
Is missing the main literature on the Sensory Organization as reported by the literature on SOT.
Response 3:
Thank you for recommending an excellent reference to improve the quality of this study. The main description of the SOT mentioned in the two papers has been added to the introduction (line 33).
Point 4:
In eye-closed condition, it is impossible to distinguish subject that increase oscillation because they are unstable or greatly confident on their ability in controlling the oscillation (the paper give ambiguous information)
Response 4:
Maintaining a posture with eyes closed is very difficult. Nevertheless, in the case of the subjects who participated in this experiment, all tasks were performed perfectly. In general eye-closed condition, it was added as a reference to the discussion as a hypothesis of a general theory. (line 202-205). Thank you.
Point 5:
The vision in eye open condition is out of control; the task condition and the instruction alter the results because they alter the context of the task, also introducing cognitive aspects.
Response 5:
We were aware of what you said and considered it during the experiment. The visual was fixed on the target in front at the eyes open condition. Added missing comments to “line 132”. And in this study, the presence or absence of vision was considered, and the difficulty according to visual control was not confirmed as a control factor. We will check the correlation with the balance ability according to the position of the vision and the blind level in a future study.
Point 6:
The pressure under the foot is under of control when you use the foam. The use of the term proprioceptive is too general in this case is preferred to refer to the pressure under the foot (the subjects simply stand).
Response 6:
It means that the perturbation caused by standing on the sponge foam has generated a difficulty condition that is given to some extent compared to the solid surface condition. The main purpose of this study is to determine the relative difference in conditions according to the presence or absence of vision. In a future study, we will implement and check the conditions that can give quantitative perturbation to the prorioceptive, which has the function of recognizing the actual pressure and movement position. This is presented as a limitation at the end of the discussion.
Point 7:
The use of the term “tasks and variables” do not seem appropriate. The task is unique: standing. The contexts are changing (Solid surface, Foam1 and Foam2). One task and three contexts.
Response 7:
It has been confirmed that “three ground conditions”, “standing posture”, are not “tasks”. It seems appropriate to simply define this as a “condition”. Therefore, all “tasks” used in surface condition and standing posture tasks have been deleted.
Point 8:
Without knowing the head and body displacements it is impossible to infers on the role of the vestibular system, please see other relevant references.
Response 8:
The difficulty of the vestibular in this study cannot be controlled, the analysis results are focused on the presence or absence of visual condition. However, in the condition of the foam, it was assumed that small surface difficulties were given to the vestibular and proprioceptive compared to the solid surface. Through future research, we will implement the conditions in which the inclination of the actual vestibular function is possible, and measure the displacement of the head and body so that the role of the vestibular system can be inferred. Thank you.
Point 9:
The perturbation induced by the alteration of the surface of support can drastically change the body frame of reference for maintain the equilibrium. In your experiment is out of control.
Response 9:
Experimental participants in this study did not deviate significantly from the reference coordinates and maintained a relatively normal standing posture. That is, there was no change in the position of the foot, and body posture. Thank you.
Point 10:
The paper need data integration and substantial revision: of the introduction integrating the proper literature, of the results integrating data, and of the discussion avoiding inference on the vestibular system on prorpiception and clearly defining all the limitations of this paper.
Response 10:
References have been added to the various comment. And, the conditions for the influence of more detailed sensory functions were limited in the experimental equipment and environment to be reflected in the design of this study. Inferences about vestibular function and proprioception other than visual function were minimized, and this was added as a limitation of this study. Thank you for letting me know about the main theories of balance ability, which I lack in much while conducting this study. In a follow-up study, we will consider that measurement and evaluation under more detailed experimental conditions can be performed. Thank you.
Round 2
Reviewer 1 Report
Dear Authors,
thank you very much for sending corrected article: Sensory Interaction and Evaluation of the Balancing Ability of Elderly using a Simplified Force Plate System. In my opinion, authors corrected it according my suggestions.
Author Response
(Cover letter)
Dear Reviewer 1.
Thank you for reviewing our manuscript. Thanks to your meaningful comments, we were able to improve the quality of this manuscript. In future research, we will derive research results that reflect your good opinions. Thanks again for the review.
Sincerely,
From Corresponding author.
Point 1: Thank you very much for sending corrected article: Sensory Interaction and Evaluation of the Balancing Ability of Elderly using a Simplified Force Plate System. In my opinion, authors corrected it according my suggestions.
Response 1: Thanks to you, we were able to improve the quality of the manuscript. thank you for the good comment.

Reviewer 3 Report
Thank you for considering the suggestions and addressing them.
Best regards
Author Response
(Cover letter)
Dear Reviewer 3.
Thank you for reviewing our manuscript. Thanks to your meaningful comments, we were able to improve the quality of this manuscript. In future research, we will derive research results that reflect your good opinions. Thanks again for the review.
Sincerely,
From Corresponding author.
Point 1: Thank you for considering the suggestions and addressing them. Best regards
Response 1: Thanks to you, we were able to improve the quality of the manuscript. thank you for the good comment.
